# Effective Invasiveness Recognition of Imbalanced Data by Semi-Automated Segmentations of Lung Nodules

**DOI:** 10.3390/biomedicines11112938

**Published:** 2023-10-30

**Authors:** Yu-Cheng Tung, Ja-Hwung Su, Yi-Wen Liao, Yeong-Chyi Lee, Bo-An Chen, Hong-Ming Huang, Jia-Jhan Jhang, Hsin-Yi Hsieh, Yu-Shun Tong, Yu-Fan Cheng, Chien-Hao Lai, Wan-Ching Chang

**Affiliations:** 1Department of Diagnostic Radiology, Kaohsiung Chang Gung Memorial Hospital and Chang Gung University College of Medicine, Kaohsiung 833, Taiwan; yctung66@gmail.com (Y.-C.T.); yu_shun_tong@hotmail.com (Y.-S.T.); prof.chengyufan@gmail.com (Y.-F.C.); o927003551@gmail.com (W.-C.C.); 2Department of Computer Science and Information Engineering, National University of Kaohsiung, Kaohsiung 811, Taiwan; a1095557@mail.nuk.edu.tw (B.-A.C.); a1095517@mail.nuk.edu.tw (H.-M.H.); a1095505@mail.nuk.edu.tw (J.-J.J.); a1095504@mail.nuk.edu.tw (H.-Y.H.); 3Department of Intelligent Commerce, National Kaohsiung University of Science and Technology, Kaohsiung 824, Taiwan; pinkwen923@gmail.com; 4Department of Information Management, Cheng Shiu University, Kaohsiung 833, Taiwan; yeongchyi@gcloud.csu.edu.tw; 5Department of Internal Medicine, Division of Pulmonary and Critical Care Medicine, Kaohsiung Chang Gung Memorial Hospital and Chang Gung University College of Medicine, Kaohsiung 833, Taiwan; sugun01@gmail.com

**Keywords:** biomedical science, lung cancer, invasiveness recognition, semi-automated segmentation, imbalance data

## Abstract

Over the past few decades, recognition of early lung cancers was researched for effective treatments. In early lung cancers, the invasiveness is an important factor for expected survival rates. Hence, how to effectively identify the invasiveness by computed tomography (CT) images became a hot topic in the field of biomedical science. Although a number of previous works were shown to be effective on this topic, there remain some problems unsettled still. First, it needs a large amount of marked data for a better prediction, but the manual cost is high. Second, the accuracy is always limited in imbalance data. To alleviate these problems, in this paper, we propose an effective CT invasiveness recognizer by semi-automated segmentation. In terms of semi-automated segmentation, it is easy for doctors to mark the nodules. Just based on one clicked pixel, a nodule object in a CT image can be marked by fusing two proposed segmentation methods, including thresholding-based morphology and deep learning-based mask region-based convolutional neural network (Mask-RCNN). For thresholding-based morphology, an initial segmentation is derived by adaptive pixel connections. Then, a mathematical morphology is performed to achieve a better segmentation. For deep learning-based mask-RCNN, the anchor is fixed by the clicked pixel to reduce the computational complexity. To incorporate advantages of both, the segmentation is switched between these two sub-methods. After segmenting the nodules, a boosting ensemble classification model with feature selection is executed to identify the invasiveness by equalized down-sampling. The extensive experimental results on a real dataset reveal that the proposed segmentation method performs better than the traditional segmentation ones, which can reach an average dice improvement of 392.3%. Additionally, the proposed ensemble classification model infers better performances than the compared method, which can reach an area under curve (AUC) improvement of 5.3% and a specificity improvement of 14.3%. Moreover, in comparison with the models with imbalance data, the improvements of AUC and specificity can reach 10.4% and 33.3%, respectively.

## 1. Introduction

### 1.1. Background

Cancer was the second top cause of deaths in 2018 according to reports by the World Health Organization (WHO) [1]. Among different cancers, lung cancer ranks first whether for male and female people. Therefore, the treatment and prevention of lung cancer attracted many researchers’ concentration in recent years. Until now, the best treatment for lung cancer was surgical removal of the tumor at the early stage. Even with surgeries, a recurrence will occur to cause death for around 20% to 30% of patients. Consequently, recent studies made on clinical medicine try to identify the recurrence factor, such as invasiveness, so as to perform necessary treatments in advance and further reduce the recurrence or mortality. Figure 1 demonstrates examples of an invasive nodule and a non-invasive nodule. In practice, the invasiveness is a critical reference that is highly related to the diagnosis, staging, treatment recommendation, and prognosis for lung cancer. With an adequate surgical resection, the patients with non-invasive nodules have 100% or near 100% disease-free survival, and those with localized invasive nodules were associated with a 5-year survival rate of 70–90%. As a result, how to effectively identify the invasiveness of early lung cancers is a very important issue researched in the field of biomedical science over the past few years. In real applications, irregular shape, solid component, and tumor size are three important considerations for identifying the invasiveness.

In general methods, the invasiveness recognition can be decomposed into two stages, namely nodule segmentation and invasiveness classification. For nodule segmentation, few recent works are very successful in partitioning the nodule automatically and precisely. This is because a precise partition needs big training data and the manual annotation cost is relatively high. Further, the invasiveness classification is not easy to pursue without precise partitions. In addition, another difficulty for traditional classifiers is to conduct an effective classification model as facing the imbalanced invasiveness data. In the imbalance data, majority samples will highly dominate the prediction, leading to biased results, such as the result with a high accuracy and a low specificity. To aim at these issues, in this paper, an effective method for recognizing the invasiveness of lung nodules is proposed via a semi-automated segmentation. The major intent behind the semi-automated segmentation is to provide the doctors with an easy way to mark the lung nodules more precisely. Just by clicking the target, the proposed method will segment the nodule automatically. Afterwards, the proposed classifier recognizes the invasiveness by boosting ensemble learning. On the whole, the main contribution can be summarized into the following aspects;

I.For technique, it further consists of two following sub-contributions, related to segmentation and classification, respectively.
(1).In terms of semi-automated segmentation, a hybrid segmentation is proposed by fusing thresholding-based morphology and deep learning-based mask-RCNN. Basically, the thresholding-based morphology is the one with statistical thresholding and mathematical shaping, while the deep learning-based mask-RCNN is a region-based convolutional neural network with a fixed anchor. Finally, the better segmentation is derived by switching them.(2).In terms of invasiveness classification, a boosting ensemble classifier is constructed by equalized down-sampling (called BEED). Especially for imbalance data, the equalized down-sampling generates multiple balanced models, and then a group decision is performed to effectively recognize the invasiveness of early lung cancers.
II.For novelty, most existing real systems mark the tumors as an initial segmentation by fully supervised learning. Then, it still needs to revise the segmentation. Otherwise, without initially automated segmentations, the manual cost is very high. These problems motivate us to conduct a semi-automated segmentation for a convenient usage. In addition to usage convenience, the semi-automated method is more effective than the fully automated ones because it employs click information to achieve a more accurate segmentation.III.For application, the proposed semi-automated segmentation satisfies the real need of generating a massive training dataset for deep learning. Additionally, the proposed invasiveness recognition can be materialized in real medical systems for effective treatments.IV.For extension, the proposed ideas of semi-automated segmentation and equalized down-sampling can be extended to other medical fields also with imbalance data such as liver tumor, brain tumor, and so on.

To realize the contributed effectiveness, the proposed methods were evaluated by numerous experiments. The experimental results deliver two aspects. First, the proposed semi-automated segmentation performs more effectively than the compared methods, measured by precision, recall, F-measure, and dice. Second, the invasiveness classification integrating equalized down-sampling and boosting ensemble learning is more robust than the competitors in terms of accuracy, sensitivity, specificity, and AUC. Overall, the semi-automated segmentation can be viewed as a support for the invasiveness classification. Technically, the imbalance problem is alleviated. In usage, the ground truth is easy to generate. In practical terms, a better treatment will be recommended. 

The remainder of this paper is organized as follows: A systematic review of past studies is provided in Section 2. In Section 3, the proposed methods of semi-automated segmentation and invasiveness recognition are presented in detail. The empirical study and the research limitation are described in Section 4 and Section 5, respectively. Finally, the conclusions are given in Section 6. 

### 1.2. Related Work

So far, there is much past literature made on biomedical science, referring to a set of algorithms learning from medical data on risk assessment, disease recognition, and treatment recommendation. To aim at the issue of disease recognition, this paper presents an approach for recognizing the invasiveness of lung nodules by a semi-automated segmentation. Because the core functions are nodule segmentation and invasiveness recognition, the review of related works is classified by three categories, namely deep learning on object segmentation, biomedical image recognition, and invasiveness recognition of lung nodules.

#### 1.2.1. Deep Learning on Object Segmentation

In general, image segmentation is sequentially proposed by two types, namely two-stage segmentation and one-stage segmentation. For two-stage segmentation, selective search [2] is the earliest one that performed hierarchical grouping to segment the objects. Afterwards, numerous related works are devoted on object detection. OverFeat [3] fused recognition, localization, and detection by CNN. Then, R-CNN [4], Fast R-CNN [5], and Faster R-CNN [6] were sequentially proposed to refine the two-stage object detection. In addition to object detection, the other paradigms in two-stage segmentation are semantic segmentation and instance segmentation. FCN [7] focused the semantic segmentation using a region-based fully convolutional network. Mask R-CNN [8] combined Faster R-CNN and FCN to achieve instance segmentation. For one-stage segmentation, based on YOLO [9], several extended versions with respect to YOLOv2 to YOLOv8 [10,11,12,13,14,15,16] were proposed to improve the segmentation performances in terms of effectiveness and efficiency.

#### 1.2.2. Biomedical Image Recognition and Segmentation

Based on deep learning, many state-of-the-arts methods [17,18,19,20] were shown to be effective on biomedical image recognition and segmentation. For image recognition, Tung et al. [21] made an empirical study to detect the scaphoid fracture. Vankdothu et al. [22] provided a brain tumor recognition for magnetic resonance imaging (MRI) images by integrating K-means and recurrent convolutional neural networks. Hart et al. [23] employed CNN to recognize the melanocytic lesions in selected whole-slide images. For image segmentation, U-net is the base proposed by Ronneberger et al. [24]. Zhou et al. [25] extended U-net as U-net++, connecting multiple U-nets. Roy et al. [26] added the squeeze and excitation (SE) module into U-net to enhance the segmentation quality. Oktay et al. [27] presented an extension of U-net called attention U-net, integrating the proposed attention gate (AG) model. Further, Ni et al. [28] enhanced the attention U-net as a residual attention U-net. Saumiya et al. [29] modified the residual U-net by residual deformable convolutions and a split graph network with convolutional spatial and channel features.

#### 1.2.3. Invasiveness Recognition of Lung Nodules

Because tumor invasiveness is a very important guidance for an effective treatment plan, how to effectively recognize the invasiveness is a hot research topic. For this purpose, many biomedical works were proposed by machine learning [30]. Qiu et al. [31] compared morphological and radiomics features for distinguishing invasive adenocarcinomas by feature selection such as the chi-square test, F1, and LASSO (least absolute shrinkage and selection operator). Kao et al. [32] aimed at invasiveness of the lung pure ground-glass nodules (PGGNs) by a radiomics prediction model based on forward sequential selection and logistic regression. Sun et al. [33] provided multivariate logistic regression analysis for PGGNs with LASSO. Song et al. [34] compared the results of using logistic regression (LR), extra trees (ET), and a gradient boosting decision tree (GBDT) with selected radiomics features. 

## 2. Materials and Methods

### 2.1. Materials

In the experiments, the data are composed of two sets used for three evaluations, namely lung segmentation, nodule segmentation, and invasiveness recognition. For lung segmentation, the data were downloaded from the kaggle competition, namely Data Science Bowl 2017 [35]. In these data, there are 50 patients with around 6500 images, which were randomly split into 5 sets for a five-fold cross validation. For nodule segmentation, the data came from the Departments of Diagnostic Radiology and Surgery, Kaohsiung Chang Gung Memorial Hospital (called KCGMH), Taiwan, containing 180 patients with 1819 images. Further, nodules of each image were marked by radiologists through the proposed online marking system. Among 180 patients, 35 patients with 326 images were randomly selected for testing and the others served for training. For invasiveness recognition, 190 nodules were selected from 180 patients of KCGMH data. A total of 160 nodules were invasive (called positive) and 30 nodules were non-invasive (called negative). A random five-fold cross validation was also executed for this evaluation. Note that the major reason for using two different sets in the experiments is interpreted by two points. First, the lung segmentation is the preprocessing of thresholding-based nodule segmentation, but it needs much effort to mark the training lungs. To save the effort, we tried to use the existing kaggle data instead of KCGMH data for training a lung segmentation model. Because the final nodule segmentation was effective, we did not use the KCGMH data to train the lung segmentation model. That is, the lung segmentation was completed finally by the kaggle training model. Yet, it leaves a future issue to be investigated for the effectiveness if using the KCGMH data. Second, on the contrary, the kaggle data are not with the invasiveness information. Therefore, they cannot be used for invasiveness recognition. For these considerations, the kaggle data were used for lung segmentation, and the KCGMH data were used for nodule segmentation and invasiveness recognition in this paper.

### 2.2. Methods

#### 2.2.1. Overview of the Proposed Approach

To achieve the contributions mentioned above, in this paper, an effective method for classifying the invasiveness of lung cancers is proposed by using a semi-automated segmentation. Figure 2 shows the framework of the proposed method, including offline preprocessing and online recognition. 

I.Offline preprocessing: In this stage, lungs are partitioned from the known computed tomography (CT) images first. Next, the necessary components are generated for online recognition, including adaptive threshold, the anchor-fixed Mask-RCNN model, and invasiveness recognition model. For the adaptive threshold, it is determined by two statistical regressions. For Mask-RCNN, it is trained with a fixed anchor. For the invasiveness recognition model, the features are extracted and filtered first. Then, a set of balanced classification models is trained by equalized down-sampling.II.Online recognition: If the offline preprocessing is completed, the online recognition starts with a submission of unknown CT images. Next, the user will click the target nodules. Then, the system attempts to segment the nodules from unknown images by thresholding. If the result is null, the segmentation will be finished by Mask-RCNN. Finally, based on the segmented nodules, the invasiveness will be recognized by the boosting an ensemble classification model called BEED.

#### 2.2.2. Lung Segmentation

This is an essential operation for offline preprocessing or online recognition. In this operation, the kernel function is an extension of the well-known Unet [24], namely SeResUnet [36]. As shown in Figure 3, it is a symmetry network consisting of a four-level encoder and a four-level decoder. At each level, the output of the encoder will be concatenated to the input of the decoder. The additional core ideas to Unet are attention mechanism and squeeze-and-excitation block, which can be referred to as Attention Unet [27] and SeNet [26], respectively. Figure 3 shows the architecture of SeResUnet where the squeeze-and-excitation block (named se_block) and the attention mechanism are embedded in the encoder and decoder, respectively. 

#### 2.2.3. Offline Preprocessing

##### Determining the Threshold Formula

In the proposed method, the first segmentation component is the thresholding-based segmentation with an adaptive threshold. This is because the difference between the nodule and non-nodule is not easy to discriminate. To overcome this problem, a formula to determine the adaptive threshold is necessary. In this stage, the threshold formula is derived by five steps. First, the CT-stored values are transformed into Hounsfield units (called hu in this paper) from the known nodules by Equation (1): *Hu* = *B* + *M* × *SV*,(1)
where *Hu* denotes the Hounsfield unit, *B* denotes the rescale intercept, *M* denotes the rescale slope, and *SV* denotes the CT stored value. Second, for each nodule, the average and standard deviation of *Hu*s are calculated. Third, a linear function for online estimating the average of unknown nodule hu values is approximated. Equation (2) is the linear function with two coefficients, which can be defined as: *Avg* =β_0_ +β_1_ × *Start*,(2)
where β_0_ and β_1_ are coefficients, *Avg* stands for the estimated average, and *Start* stands for the pixel (hu) clicked by the user. Based on the known averages and user clicks, the β_0_ and β_1_ are approximated as −173.34 and 0.71, respectively, by a simple linear regression in this paper. Fourth, a function for online estimating the unknown standard deviation *Std* is derived, which is defined as:*Std* = *Regression*(*Start*, *Avg*),(3)
where *Regression*(*,*) is the multiple regression model trained by known *Starts*, averages, and standard deviations. Fifth, the threshold formula is thereby derived, which is defined as: *Threshold* = *Avg* − *α* × *Std*,(4)
where *Avg* is calculated by Equation (2), *Std* is calculated by Equation (3), and *α* is the weight of *Std*. The larger the *α*, and the smaller the threshold, the larger the segmented area. Finally, three results with respect to simple linear formula, regression model, and threshold formula are ready for online thresholding-based segmentation. Note that the *α* is set as 2 in this paper.

##### Training the Anchor-Fixed Mask-RCNN

The other segmentation component in addition to thresholding is the revised Mask-RCNN. In fact, it is more efficient and effective than the original Mask-RCNN because the anchor is fixed by the clicked pixel. That is, without scanning the whole image, the search space is located around the fixed anchor. Figure 4 shows an example of anchor-fixed Mask-RCNN, which indicates that the anchor is fixed to generate multiple potential rgion of interests (ROIs). Thus, it is more precise and fast than the original Mask-RCNN, which can be referred to in the experimental results in Section 4. 

Figure 5 shows the framework of Mask-RCNN [7], consisting of fully convolutional networks (FCN) and Faster R-CNN (region-based CNN). In the original Mask-RCNN, the images are processed into ROIs with a size of 7 × 7 through the convolutional neural network (CNN), region proposal network (RPN), and region of interest align (ROI Align). Then, the FCN generates the mask, and the Faster R-CNN performs the class prediction (Class pred.) and bounding box prediction (BBox pred.). Finally, the objects will be segmented semantically by fusing results of FCN and Faster R-CNN. Because it needs a number of anchors to predict, the computational cost is high. To deal with this problem, in this paper, we modify the Mask-RCNN as a revised Mask-RCNN from anchor-free to anchor-fixed. This is because the object information is given by the user click, so as to reduce the prediction cost without scanning all anchors. In this paper, for each feature map, 3 × 3 anchors around the clicked pixel will be calculated. The complexity of revised Mask-RCNN over that of the original one reaches 21824/45 ≈ 473 times. 

##### Training the Invasiveness Recognition Model

On the basis of ground truths, the proposed invasiveness recognition model is constructed by a set of balanced models, which can be viewed as an ensemble model. Because the minority data are much smaller than the majority data, the primary intent of the proposed method is to cover all potential cases, where each case combines the minority set with different majority subsets. Actually, it can also be regarded as a group decision with ensemble learning. Oriented from this idea, the majority data are randomly divided several equalized subsets and each subset is equivalent to the minority data size. Such processing is called equalized down-sampling in this paper, which can be defined as that; assume there are *m* majority elements and *n* minority elements. There will be (*m*/*n*) subsets generated where the ratio of majority to minority is *n*:*n*. After down-sampling, this stage will be decomposed into several steps. First, the radiomics features [37,38] are extracted from the nodules. Second, the feature selection will be performed to approximate nearly optimal features. Third, a set of collaborative classifiers will be trained for effectiveness assessments. Fourth, each classifier is assigned a weight. The procedure of assigning weights is shown in Figure 6. Lines 1–7 depict that the AUC of each classifier is derived by classifying the other training sets. Lines 8–9 depict that the maximum and minimum AUCs are determined. Lines 10–13 depict that the classifier weight is derived by the minimum to maximum normalization. Finally, a number of weighted classification models are constructed for online recognition.

#### 2.2.4. Online Recognition

The online recognition is triggered with a submission of unknown images. For each image, the lung is segmented first. Thereupon, the nodule segmentation and invasiveness recognition are executed sequentially. The details of online recognition will be presented in the following subsections.

##### Thresholding-Based Morphology for Semi-Automated Segmentation

This is the first stage for nodule segmentation, which is based on the segmented lung and a user click named *Start*. That is, the search space in this stage is limited in the lung and the click. As shown in Figure 7, this stage can be decomposed into two phases, namely thresholding and morphology. In the first phase, the hu features are extracted from the image first. Next, the user click is input to Equation (2) and the average is thereby derived. Then, with the user click and estimated average, the standard deviation is calculated by Equation (3). Finally, based on the estimated average and standard deviation, the threshold is approximated by Equation (4). According to the approximated threshold and *Start*, the initial segmentation is derived by binary thresholding. The main idea of binary thresholding is to perform the pixel connection under the threshold constraint. In this method, the binary thresholding uses eight neighbors surrounding the central cell as the connection shape. Although the threshold is adaptive, there still exist possible connections between the nodule and non-nodule. Figure 8b shows an example of this problem. Therefore, a set of morphology operations is necessary. First, the erodge is performed to split the nodule and non-nodule. Afterwards, the nosie removal and dilate are performed to reshape the result. In this paper, the noises indicate the areas do not include the user click. Figure 8a is an example of the input image, and Figure 8b–e shows the related results of all operations. In the morphology, the kernel size is 3 × 3 and the number of iterations is 1.

##### Deep Learning-Based Mask-RCNN for Semi-Automated Segmentation

In the proposed segmentation, Mask-RCNN is proposed as a complementary solution while the thresholding-based morphology cannot find the nodules; that is, no result is segmented. To make Mask-RCNN more effective and efficient, the Mask-RCNN is modified as an anchor-fixed version, called revised Mask-RCNN. Even fixing the anchor, an issue to cope with is the determination of candidate segmentation results. For this issue, the final segmentation for revised Mask-RCNN is the one with the maximum overlap with all segmentations results. Figure 9 is an example of determining the final results of revised Mask-RCNN. In this example, there are three candidate results, including A, B, and C. Given that the union of intersections is U = ∪{D, E, F, G}, the final result is B because the overlap between B and U is maximum. 

##### Invasiveness Recognition

As shown in Figure 2, the workflow comprises steps of radiomics features extraction, feature filtering, and classifying. Based on the selected features, the collaborative models classify the unknown nodule. Next, the weighted positive results are summed. Finally, if the summed result exceeds the criteria, it will be predicted as positive. Otherwise, it will be recognized as negative. Thereupon, the classification result will be a treatment consideration for the doctor. Note that the criterion in this paper is 0.5.

## 3. Results

After presenting the proposed method in the preceding section, what we want to show next is the justification of contributed effectiveness. Because the proposed method is composed of two main stages, namely semi-automated segmentation and invasiveness recognition, the experiments were made on these two topics. In this section, the details of experimental settings and empirical results will be shown by a number of comparative evaluations. At last, an insightful discussion will be lifted. Via the experimental results, the contributions on two topics can be clarified clearly.

### 3.1. Experimental Settings

Two types of experimental measures are employed for segmentation and recognition. One is measuring the segmentation quality based on spatial intersections, such as *Dice*, *Precision*, *Recall,* and *F*-measure, which can be defined as:(5)$$Dice=\frac{2\;\times \;|Predicted\cap Truth|}{\left|Predicted|+|Truth\right|}$$
(6)$$Precision=\frac{\left|Correct\right|}{\left|Predicted\right|}$$
(7)$$Recall=\frac{\left|Correct\right|}{\left|Truth\right|},\;\mathrm{and}$$
(8)$$F-measure=\frac{2\left(Precision\;*\;Recall\right)}{Precision+Recall}$$
where *Predicted* denotes the set of segmentations, *Correct* denotes the set of correctly segmented ones, and *Truth* denotes the set of ground truths. The other experimental measures were employed for validating the recognition quality based on a confusion matrix. Basically, this matrix includes four elements, namely true positive (*TP*), false positive (*FP*), false negative (*FN*), and true negative (*TN*), representing the successful predictions and failed predictions for positives and negatives, respectively. According to this matrix, measures of *Accuracy*, *Sensitivity,* and *Specificity* are defined as:(9)$$Accuracy=\frac{TP+TN}{TP+TN+FP+FN}$$
(10)$$Sensitivity=\frac{TP}{TP+FN},\;\mathrm{and}$$
(11)$$Specificity=\frac{TN}{TN+FP}$$

In addition to *Accuracy*, *Sensitivity,* and *Specificity*, the other recognition measure is *AUC*, indicating the area under the curve of receiver operating characteristic (ROC), presenting the classification performance under different thresholds of true positive rates and false positive rates. In general, *Accuracy* represents the overall prediction quality and *Precision* represents the successful rates of prediction results. In contrast, *Sensitivity* and *Specificity* represent true positive rates and true negative rates, respectively, for ground truths, which are sensitive to balance data. 

### 3.2. Experiments on Semi-Automated Segmentation

#### 3.2.1. Results of Lung Segmentation

As we can recall from Figure 2, lung segmentation is the fundamental component before thresholding-based morphology. Table 1 depicts the results of a five-fold cross validation by using SeResUnet. On average, the dice reaches 98.6% with an insignificant standard deviation of 0.168, representing a robust recognition result. Accordingly, the best one in these five models was selected as the lung segmentation model in the succeeding binary thresholding segmentation. In this model, the related parameters of epoch, batch size, and learning rate are 100, 2, and 0.0001, respectively.

#### 3.2.2. Ablation Study 

In the semi-automated segmentation, it comprises components of thresholding-based morphology (termed TM in the experiments) and deep learning Mask-RCNN (termed MR in the experiments). Hence, clarifications for the impacts of individual components are necessary. Figure 10 shows the ablation study, presenting that the segmentation fusing two components achieves the best dice in contrast to individual ones. Figure 11 is the further analysis, showing that around 82.2% of ground truths can be detected and 75.4% of predictions are correct. Note that in this evaluation, the parameters of Mask-RCNN for epoch, batch size, and learning rate are 30, 1, and 0.001, respectively.

#### 3.2.3. Comparisons with Existing Semi-Automated Segmentation Methods 

From the ablation study above, we can know that the proposed thresholding-based morphology and deep learning Mask-RCNN are complementary components for a better fusion. The final issue to address for semi-automated segmentation is: what if it is compared to that of other existing methods? For this issue, a number of existing methods were compared with the proposed fusion method (named SSTM, semi-automated segmentation of fusing TM and MR). Table 2 shows the compared methods, including five existing methods. In these methods, level set is a segmentation that shapes the object by numerical calculations on a Cartesian grid. Static threshold fixes the optimal threshold by experiences, which was set as −750 in this paper. The other three methods dynamically set the thresholds by mean, Gaussian, and standard deviation of objects, respectively. Note that all compared methods are based on the pixel named *Start* clicked by the user.

Figure 12 shows the comparison among the compared methods in terms of dice. This comparison delivers some aspects. First, the level set is better than binary thresholding methods. Second, the method with a static threshold is much better than that with adaptive thresholds. The potential reason is that the feature difference between nodule and non-nodule is not derived by these methods. Third, in contrast to the compared methods, the proposed method SSTM takes advantages of both TM and MR to achieve a better result. Here, the main advantage of TM is to approximate a robust threshold by two regression models, while MR is superior to distinguish the feature difference by region-based CNN.

#### 3.2.4. Illustrative Examples of Segmentation Results

Figure 13 shows illustrative examples of better results made by thresholding-based morphology and deep learning-based Mask-RCNN, respectively. In Figure 13b, deep learning-based Mask-RCNN is better than thresholding-based morphology in segmenting the complicated nodule. This is because the thresholding-based morphology cannot deal with the nodules out of the lung. In this case, the pixels of nodules and non-nodules will be connected in the thresholding-based morphology. On the contrary, deep learning-based Mask-RCNN searches the nodules without limiting the search space in the lung. It is good at feature filtering while recognizing complicated nodules. This is why the segmentation switches between these two methods, which can be evidenced by experiments.

### 3.3. Experiments on Invasiveness Recognition

#### 3.3.1. Effectiveness of Feature Selections for Compared Classifiers without Data Balancing

In this subsection, three feature selection methods, namely the chi-squared test, ANOVA [43], and information gain and Pearson correlation are evaluated with six classifiers. Table 3 shows the compared methods. Figure 14, Figure 15, Figure 16 and Figure 17 demonstrate the resulting AUCs under different features selected, which can be summarized into a set of points. First, the chi-squared test and ANOVA perform pretty close, which is slightly better than information gain and slightly worse than the Pearson correlation. Second, four feature selection methods do not bring out an obvious improvement over that using the full features. Third, on average, the best settings of the chi-squared test, ANOVA, information gain, and Pearson correlation are 1200, 900, 1200, and 900, respectively, but the differences are very small. Note that the evaluations in this subsection are made without data balancing.

#### 3.3.2. Comparisons of Balancing and Unbalancing Methods for Selected Classifiers

On the basis of the feature selection performances, RF, LDA, and XGBoost with 300 information gain features, full features, and 1200 ANOVA features, respectively, were evaluated further. In this evaluation, the compared methods contain the synthesized minority oversampling technique (SMOTE) [44,45] and imbalanced, where SMOTE is a well-known up-sampling method and imbalanced indicates no balancing operation is performed for classifiers. 

Table 4 elaborates the comparative result. From this result, we can know that, first, the best accuracy, AUC, sensitivity, and specificity are derived by imbalanced, BEED, and imbalanced and BEED, respectively. Second, RF plays the role of better kernel classifier in contrast to the other ones. Third, the proposed BEED is more promising than SMOTE. Fourth, although imbalanced performs better than BEED, it is weaker in terms of AUC and specificity. In this paper, AUC and specificity are the main attention in this paper. That is, the experimental results reveal that the goal of dealing with imbalance data is achieved by the proposed method. Finally, the recommended model is Random Forest with 300 information gain features. 

## 4. Discussion

In the above experiments, the technical contributions on semi-automated segmentation and boosting ensemble classification were examined. Yet, there actually remain a number of issues to clarify further. In this subsection, an insightful discussion will be provided for a more solid concern.

I.For the mathematical morphology, a further concern needs to be clarified here. In the morphology, the object is reshaped by an erode and a dilate. The primary idea is to delete the noises and to restore the original shape. However, a potential question might thereby be caused: what if varying the numbers of erodes or dilates? Figure 18 shows the answer that the morphology fusing of one erode and one dilate is better than the others. This is because two dilates are too many for one erode. In contrast, for two erodes, two dilates recover the deleted but not complete. Additionally, the morphology with one erode and one dilate is cheaper than the others.II.In Equation (4), the parameter *α* determines the threshold highly related to the initial segmented area in binary thresholding. A small threshold might lead to a high recall and low precision. Otherwise, high precision and a low recall might be caused. Therefore, an extended issue for the impact of *α* is investigated here. Figure 19 shows the effectiveness of the proposed method under different settings of *α* in terms of precision, recall, and dice, which reaches the best dice as *α* = 2, with a balance between precisions and recalls. It is obvious that the recall increases as *α* increases. This is because the segmented area increases simultaneously. However, a larger *α* will cause a lower precision. This is why the *α* is set as 2 in this paper.III.The goal of semi-automated segmentation is to provide the doctors with an efficient and effective tool for marking the nodules. Actually, most existing marking systems perform the fully automated segmentation as an initial mark. Then, it is revised by the doctor. Hence, a potential question for effectiveness differences of the proposed semi-automated segmentation and fully automated ones needs to be replied. For this question, three recent fully automated segmentation methods, including Mask-RCNN [7], Unet [24], and SeResUnet [34] were compared with the proposed method SSTM. Figure 20 reveals that the proposed SSTM achieves much better dice than the fully automated methods, reaching a dice improvement of 392.3%. The first potential reason is that the training data for the compared methods are not enough. Second, additional click information is very helpful to segmentation. In summary, this result says that the proposed idea is robust if facing small data. Moreover, it is easy and cheap. Note that all methods were executed with the same experimental settings.IV.The final issue to discuss in this paper is the scalability of the proposed methods, showing the capability of handling the data size variation. It can be interpreted by two categories, namely nodule segmentation and invasiveness recognition. Whether for nodule segmentation or invasiveness recognition, the training data sizes were set from 70% to 90% in this evaluation. Figure 21 and Figure 22 show the related results in terms of dice, AUCs, accuracies, sensitivities, and specificities, respectively. Although the larger training data sizes for all measures achieve the better results, the differences are not significant. It delivers an aspect that the proposed method is not very sensitive to the training data size.

## 5. Research Limitation

Although the proposed methods are demonstrated to be effective by numerous evaluations, there still exist some limitations to address here. First, the experiments were conducted on a personal computer with Intel(R) Core(TM) i7-10700 K CPU @2.9 GHz 2.9 GHz, 16 GB RAM and NVIDIA GeForce RTX 3070 GPU, 1.73 GHz, 8 GB memory. This specification is highly related to the deep learning performance. Second, the experimental data were gathered from Kaohsiung Chang Gung Memorial Hospital, Taiwan. Third, the settings of Equations (2) and (3) are approximated by the proposed experimental data. Fourth, the data size is limited in the number of experimental data. Fifth, the image type is limited in CT. Sixth, the aim is the lung nodule.

## 6. Conclusions

Artificial intelligence in biomedical science was actually studied for a long time. Especially for advances of deep learning, a significant success in risk assessment, disease recognition, and treatment recommendation were also approached in recent years. Although there were a number of previous studies proposed on lung tumor segmentation and invasiveness recognition, the results leave room for improvement. First, the segmentation needs a high-priced manual cost for marking the data. Otherwise, the automated segmentation cannot cater to the need of high quality. Second, it is not easy to overcome the problem of imbalance data in recognizing the invasiveness. To alleviate such problems, in this paper, a boosting ensemble classifier is presented by the proposed semi-automated segmentation. In terms of semi-automated segmentation, an effective method named semi-automated segmentation of fusing TM and MR (SSTM) integrating thresholding-based morphology (TM) and deep learning Mask-RCNN (MR) is proposed. The creativity of SSTM in contrast to conventional methods can be summarized into three aspects. First, for TM, an adaptive threshold is inferred by two statistical regressions. Then, the initial segmentation is refined by morphology operations. Second, for MR, the anchor is specified to narrow the search space into the potential area. Third, the complementary results improve the performance significantly. In terms of invasiveness recognition, the proposed boosting ensemble classifier named BEED enhances the imbalanced recognition by equalized down-sampling. An empirical study made on real data demonstrates that the performance of the proposed methods is more promising than compared methods in nodule segmentation and invasiveness recognition. From the usage point of view, marking the ground truth is no longer difficult and expensive. Moreover, effective treatments can be made according to the better recognition results. In the future, there are some unsettled problems to handle. First, the threshold will be approximated adaptively by other optimizers in addition to regressions. Second, the proposed ensemble learning is limited in one classifier. In the future, different classifiers will be incorporated into an enhanced classifier. Third, to investigate the sensitivity of the proposed method for different data, we will look for international collaborations to evaluate the proposed method by global data. Fourth, the weighted loss function will be tested to address the imbalance problem. Fifth, in practical use, applying the proposed method was requested from the hospital divisions such as gastroenterology, chest, and so on. Therefore, it will be materialized into the existing biomedical system in the hospital in the future.

## Figures and Tables

**Figure 1 biomedicines-11-02938-f001:**
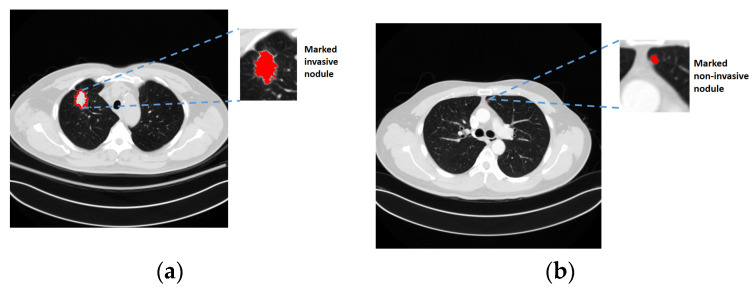
(**a**): Example of an invasive nodule; (**b**): example of a non-invasive nodule.

**Figure 2 biomedicines-11-02938-f002:**
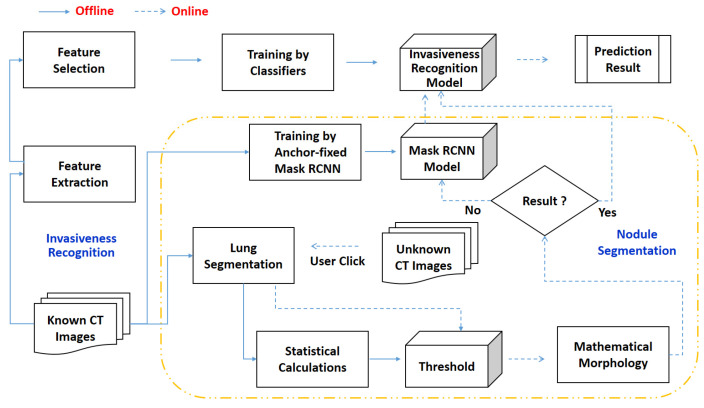
Framework of the proposed approach. The dotted-lined arrows denote the online recognition stage and the dotted yellow lines denote the Nodule Segmentation.

**Figure 3 biomedicines-11-02938-f003:**
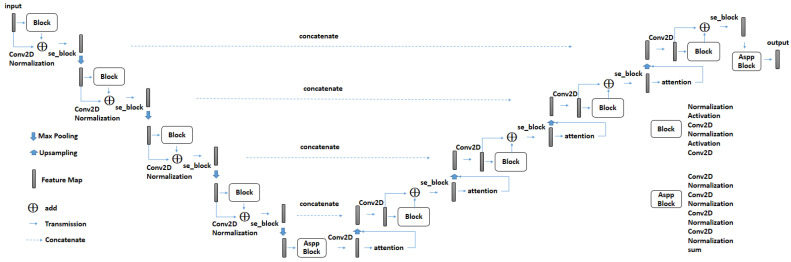
SeResUnet.

**Figure 4 biomedicines-11-02938-f004:**
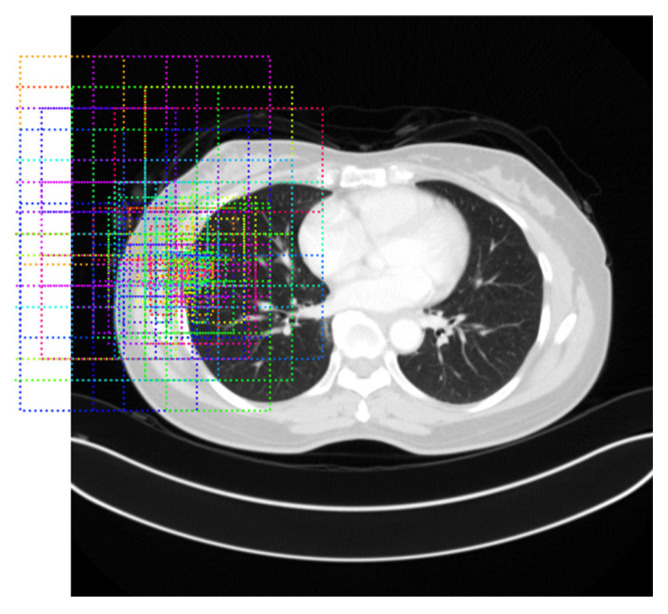
Example of performing anchor-fixed Mask-RCNN where the dotted rectangles denote the regions of interests.

**Figure 5 biomedicines-11-02938-f005:**
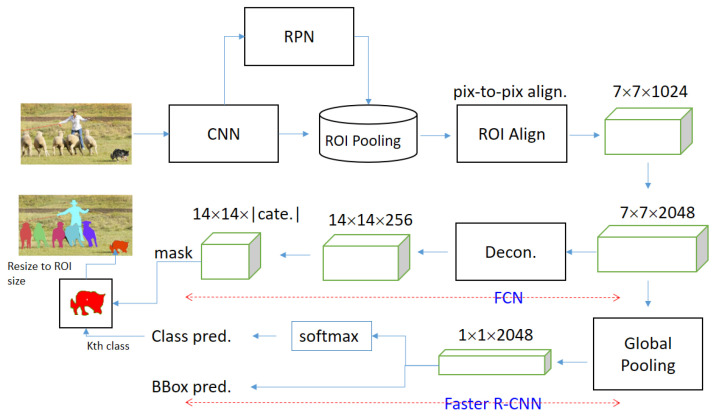
Workflow of Mask-RCNN where the red dotted-lined arrows denote ideas of fully convolutional networks and Faster R-CNN Finally, the person and animals are marked in different colours.

**Figure 6 biomedicines-11-02938-f006:**
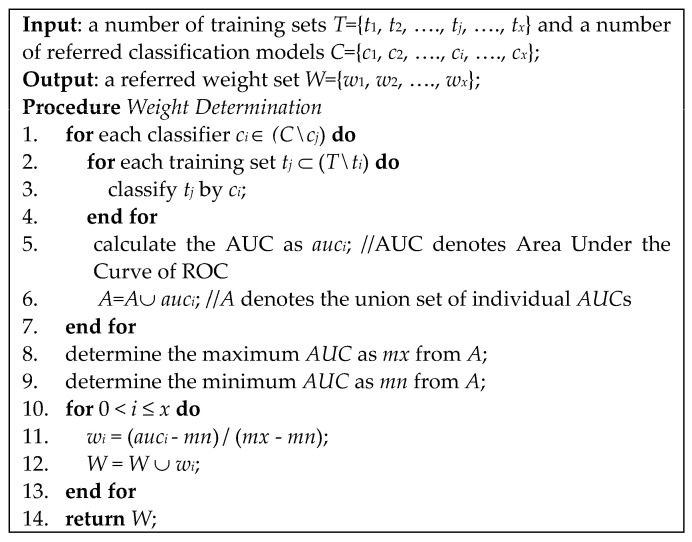
Procedure of determining the weights of collaborative classifiers.

**Figure 7 biomedicines-11-02938-f007:**
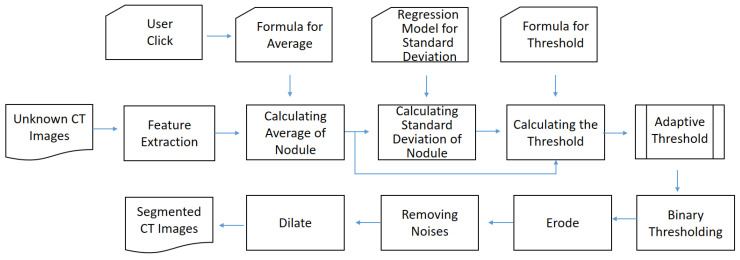
Workflow of thresholding-based morphology.

**Figure 8 biomedicines-11-02938-f008:**
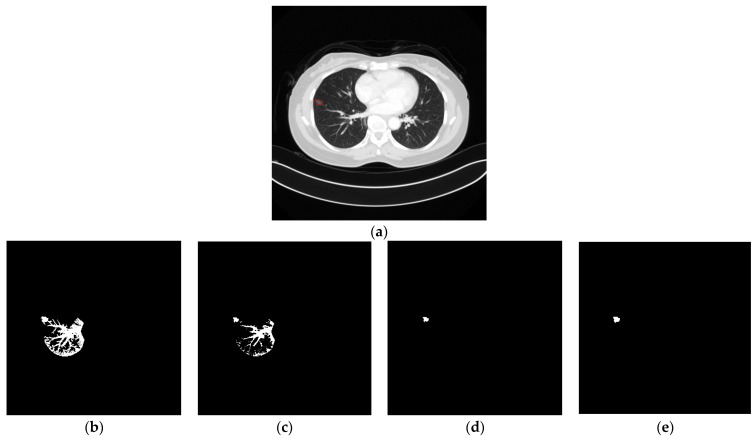
(**a**): Original image with a tumor marked in red; (**b**): result of thresholding; (**c**): result of erode; (**d**): result of nosies removal; and (**e**): result of dilate.

**Figure 9 biomedicines-11-02938-f009:**
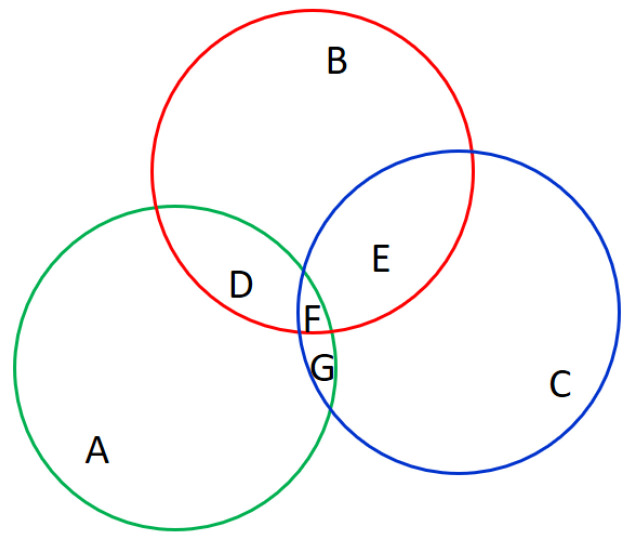
Example of determination of final results of revised Mask-RCNN where three colored circles denote three segmentation results.

**Figure 10 biomedicines-11-02938-f010:**
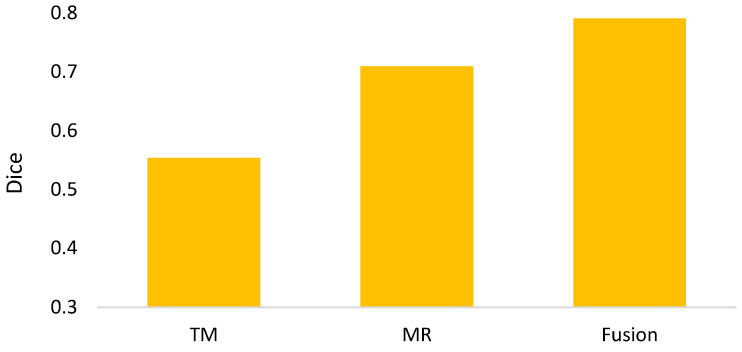
Ablation study for the semi-automated segmentation.

**Figure 11 biomedicines-11-02938-f011:**
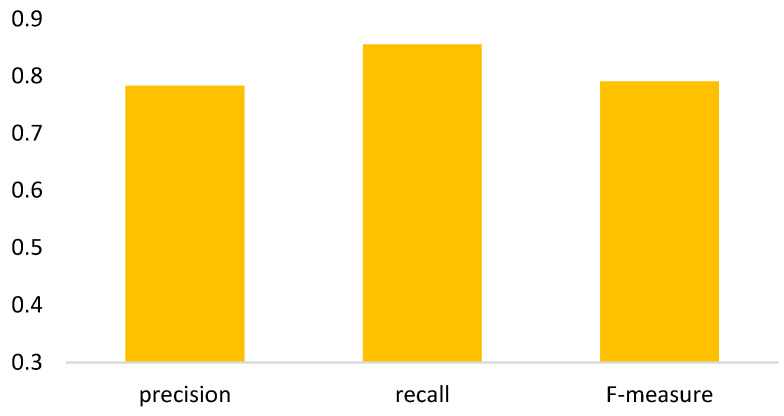
Precision, recall, and F-measure for the fusion segmentation.

**Figure 12 biomedicines-11-02938-f012:**
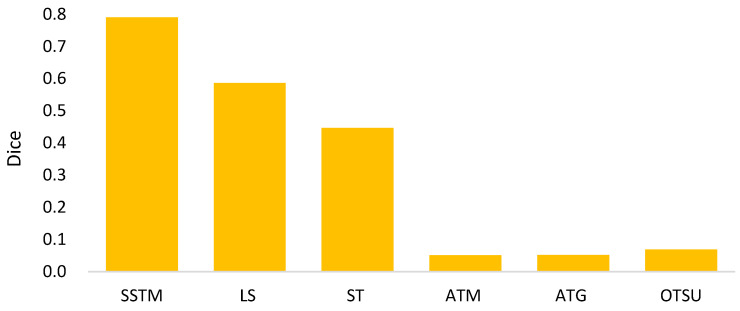
Dices of compared methods for semi-automated segmentation.

**Figure 13 biomedicines-11-02938-f013:**
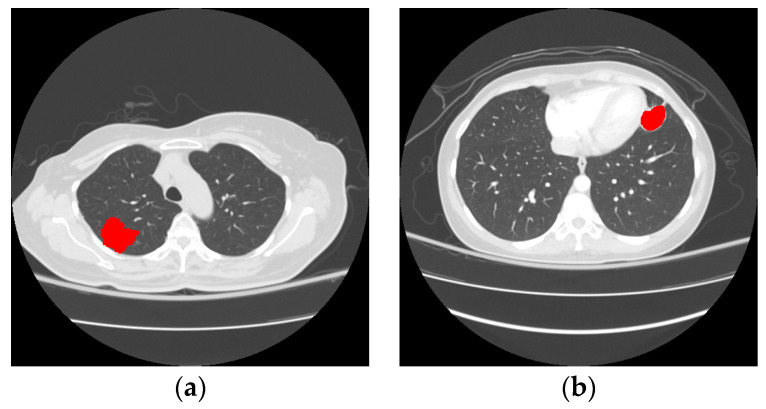
(**a**): Example of better results by thresholding-based morphology; (**b**): example of better results by deep learning-based Mask-RCNN. Note that the red areas denote the segmented nodules.

**Figure 14 biomedicines-11-02938-f014:**
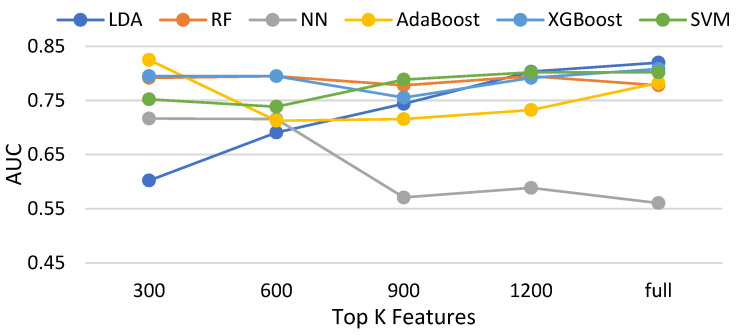
AUCs of compared classifiers using feature selection of the chi-squared test.

**Figure 15 biomedicines-11-02938-f015:**
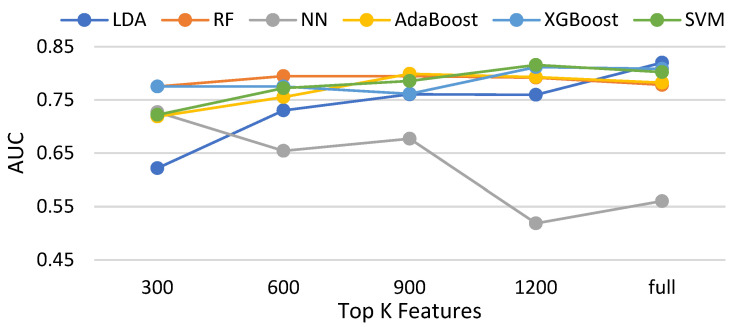
AUCs of compared classifiers using feature selection of ANOVA.

**Figure 16 biomedicines-11-02938-f016:**
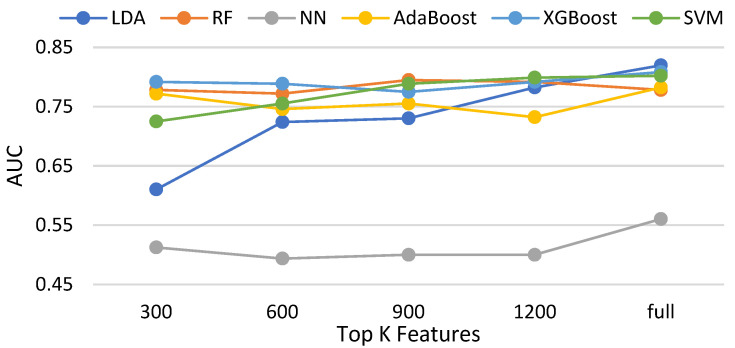
AUCs of compared classifiers using feature selection of information gain.

**Figure 17 biomedicines-11-02938-f017:**
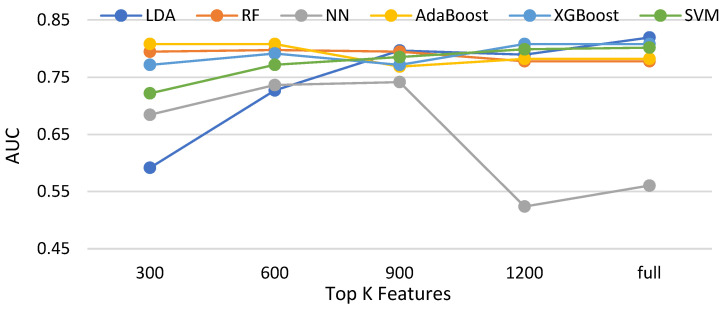
AUCs of compared classifiers using feature selection of Pearson correlation.

**Figure 18 biomedicines-11-02938-f018:**
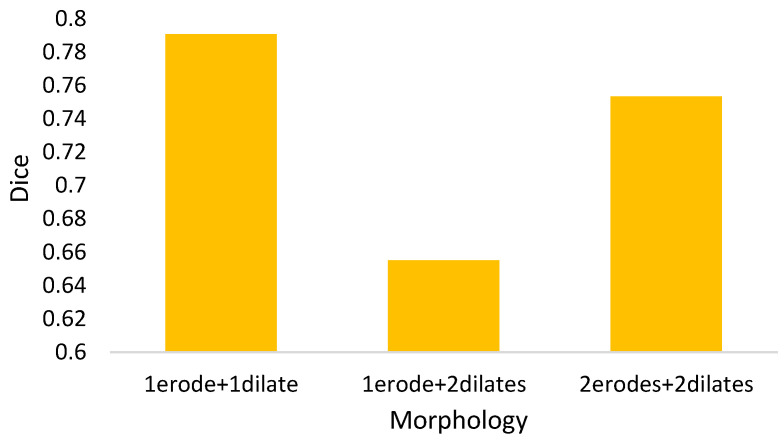
Dices for different morphology settings.

**Figure 19 biomedicines-11-02938-f019:**
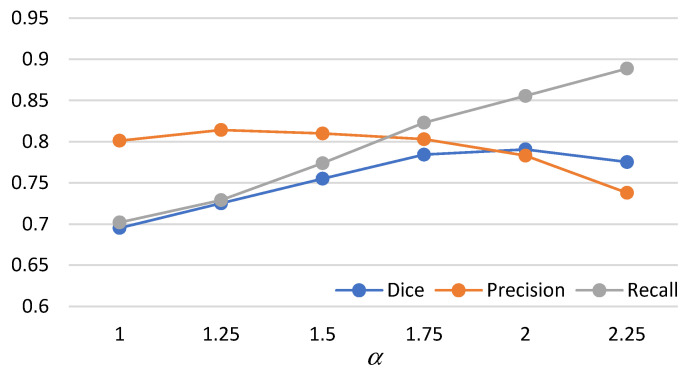
Precisions, recalls, and dices for different *α*.

**Figure 20 biomedicines-11-02938-f020:**
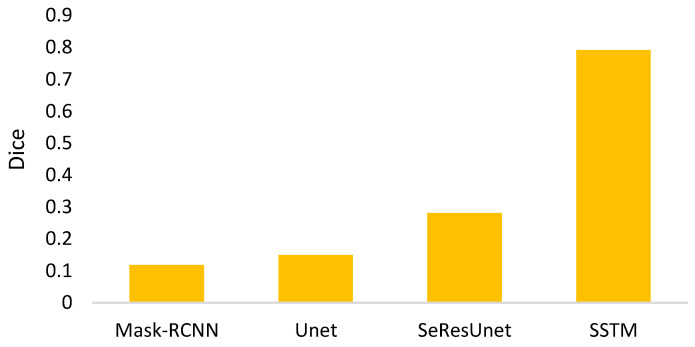
Dices of compared segmentation methods.

**Figure 21 biomedicines-11-02938-f021:**
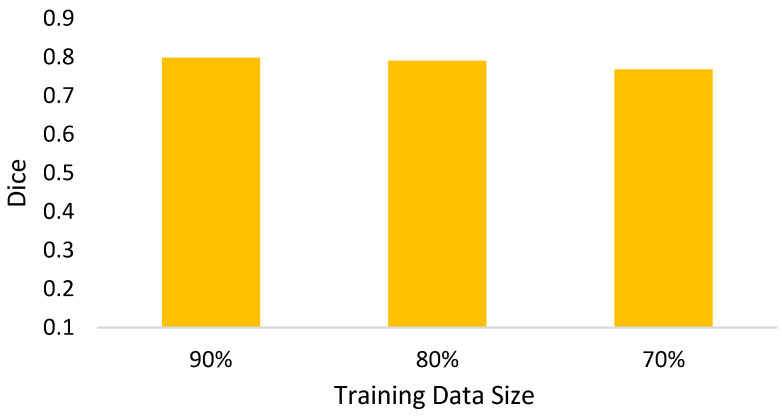
Dice of the proposed nodule segmentation for different training data sizes.

**Figure 22 biomedicines-11-02938-f022:**
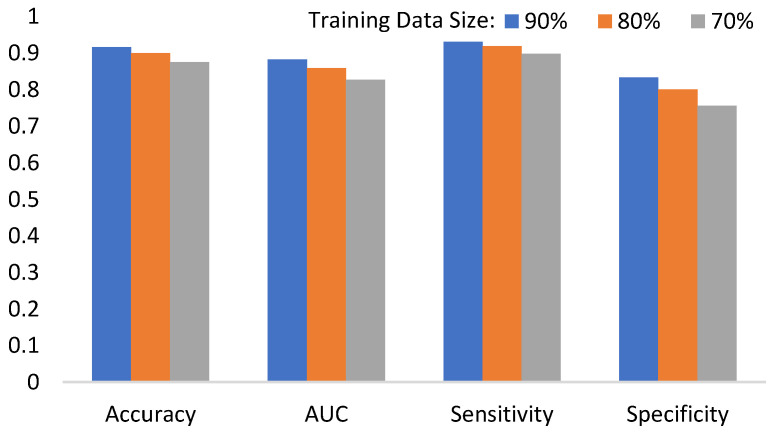
AUCs, accuracies, sensitivities, and specificities of the proposed invasiveness recognition for different training data sizes.

**Table 1 biomedicines-11-02938-t001:** Five-fold validation for lung segmentation.

Fold #	Dice	Standard Deviation
Fold 1	0.991	0.104842
Fold 2	0.982	0.241201
Fold 3	0.984	0.247245
Fold 4	0.990	0.046181
Fold 5	0.983	0.198752
Average	0.986	0.167644

# denotes the number of the fold.

**Table 2 biomedicines-11-02938-t002:** Compared methods for semi-automated segmentation.

Method	Terminology
Proposed Fusion of TM and MR	SSTM
Level-Set [39]	LS
Static Threshold	ST
Adaptive Threshold by Mean [40]	ATM
Adaptive Threshold by Gaussian [41]	ATG
Adaptive Threshold by OTSU [42]	OTSU

**Table 3 biomedicines-11-02938-t003:** Compared classifiers for invasiveness recognition.

Classifier	Terminology
Linear Discriminant Analysis	LDA
Random Forest	RF
Neural Network	NN
AdaBoost	AdaBoost
XGBoost	XGBoost
Support Vector Machine	SVM

**Table 4 biomedicines-11-02938-t004:** Effectiveness comparisons of balancing and unbalancing methods for RF, LDA, and XGBoost with specific feature selections.

		Accuracy	AUC	Sensitivity	Specificity
**RF** **(Information Gain 300)**	BEED (proposed)	0.9	0.859 *	0.919	0.8 *
	SMOTE	0.895	0.816	0.931	0.7
	Imbalanced	0.9	0.778	0.956 *	0.6
**LDA** **(Full Features)**	BEED (proposed)	0.853	0.764	0.894	0.633
	SMOTE	0.663	0.624	0.681	0.567
	Imbalanced	0.879	0.82	0.906	0.733
**XGBoost** **(ANOVA 1200)**	BEED (proposed)	0.884	0.782	0.931	0.633
	SMOTE	0.874	0.79	0.913	0.667
	Imbalanced	0.911 *	0.811	0.956 *	0.667

* denotes the best performance in the column.

## Data Availability

The patients of this study did not give written consents for their data to be shared publicly, so due to the sensitive nature of the research, supporting data are not available.

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
