# Peer review of "Effective Invasiveness Recognition of Imbalanced Data by Semi-Automated Segmentations of Lung Nodules"

_biomedicines, 2023, doi:10.3390/biomedicines11112938_

Round 1

Reviewer 1 Report

** lines 47, 48: expand to worldwide report and add references. Do not just cite Taiwan

** lines 112 – 116: poor English, rewrite

** lines 184 – 185: explain why figure 2.b is better than figure 2.a

** line 215: replace “is” by “shows” for better English

** line 254 – 255 and Eq(1): explain how you got the coefficients shown Eq(1). Did you fit a linear regression to dataset?

** lines 25: explain how you arrived at Eq(2) and why you set alpha to 2 (line 263)

** Section 4: you compare different methods of semi-automated segmentation, but you never compared semi-automated segmentation to fully-automated segmentation. Add that comparison to see how more accurate semi method than fully method

** try to find a dataset outside Taiwan to validate your approach at an international level

** needs minor English editing

Author Response

The authors are grateful for the reviewers’ helpful comments that are valuable in improving this paper. We have revised the manuscript as follows.

Revision made in accordance with comments by Reviewer No.1

  1. lines 47, 48: expand to worldwide report and add references. Do not just cite Taiwan.

Answer: Thanks for this comment. We have expanded the report from Taiwan to World Health Organization (WHO), which is presented in Section 1 (P. 2) and cited as the reference [1].  

  1. lines 112 – 116: poor English, rewrite.

Answer: Thanks for this comment. We have rewritten the descriptions in Section 1 (P. 3). 

  1. lines 184 – 185: explain why figure 2.b is better than figure 2.a.

Answer: Thanks for this comment. This is because the thresholding-based morphology cannot deal with the nodules out of the Lung. In this case, the pixels of nodules and non-nodules will be connected for thresholding-based morphology. On the contrary, DeepLearning-based Mask-RCNN searches the nodules without limiting the search space in the Lung. It is good at feature filtering while recognizing complicate nodules. This is the main reason of the segmentation switches between these two methods. For this comment, we modified the interpretation for why Figure 13-(b) is better than Figure 13-(a) if using Mask-RCNN. The modification is presented in Subsection 4.2.4 (P. 13). Note that, Figure 2 has been renamed as Figure 13 and moved to Subsection 4.2.4 in this revision. 

  1. line 215: replace “is” by “shows” for better English.

Answer: Thanks for this comment. It has been modified as the reviewer suggested. The modification is shown in Subsection 3.1 (P. 4).

  1. line 254 – 255 and Eq(1): explain how you got the coefficients shown Eq(1). Did you fit a linear regression to dataset?

Answer: Thanks for this comment. Yes, Eq. (1) is derived by a simple linear regression approximated by the proposed dataset. Actually, the coefficients should be determined dynamically by datasets. Therefore, we updated the Eq. (1) by setting the coefficients as parameters. The related description was also modified, which was shown in Subsection 3.3.1 (P. 6). Also, the related settings have been set as one of research limitations (Section 5, P. 18).

  1. lines 25: explain how you arrived at Eq(2) and why you set alpha to 2 (line 263).

Answer: Thanks for this comment. Note that, because we added an additional Equation named Eq. (2), the original Eq. (2) was renamed as Eq. (3) in this revision.

Eq. (3) is proposed for approximating the threshold. It contains three parameters, namely std, a and avg, where avg is derived by Equation (1), std is derived by Eq. (2) and a is the weight of std. For this comment, we have modified the statements in Subsection 3.3.1 (P. 6).

For a, it is set by an empirical study which is shown in Figure 19. The related analysis is shown in Subsection 4.4-II (P. 17). 

  1. Section 4: you compare different methods of semi-automated segmentation, but you never compared semi-automated segmentation to fully-automated segmentation. Add that comparison to see how more accurate semi method than fully method

Answer: Thanks for this comment. To investigate the difference between the semi-automated segmentation and fully-automated segmentation, we made a comparison between them. The result is shown in Figure 20 and the related analysis is shown in Subsection 4.4-III (P. 17). In overall, the proposed SSTM achieves a much better dice than the fully-automated methods.

  1. try to find a dataset outside Taiwan to validate your approach at an international level

Answer: Thanks for this comment. Yes, we agreed with this point. For this issue, we tried our best to search the data for lung tumors with invasiveness, but in vain. We guessed it might be limited by the ethical issues. For this concern, it was stated in the research limitations (Section 5, P. 18). Also, we added this suggestion into the future work (Section 6, P. 19).

Reviewer 2 Report

The paper seems to propose a local semi-automatic segmentation method for measurement of nodule size from CT images. Invasiveness of the nodules is classified. The paper is very long and hard to follow, due to the redundance of information. Despite the length, some important detail are missed don’t’ allowing reproducibility of results. My advice is t dramatically shorten the paper attempting to focus on the introduced novelty and better detailing the proposed methodology.

Authors seems to use three different data set to validate each step of the workflow. This kind of validation will no demonstrate the effectiveness of the whole workflow. Why do no test the whole algorithm on a single data set as the last one?

Abstract: The abstract should resume the overall paper content, including the significant results.

Introduction: How invasive/non invasive nodules are classified in the clinical practice?  By size ,signal or shape?

3. I don’t see the need o this section. Authors mix the description of the implemented algorithm with discussion about advantages of the proposed implementation. I suggest to write a more rigorous method section with a objective description of the implemented workflow. Advantages of the methos should be described I the discussion section basing on obtained results.

Figure 2/3 anticipates results of algorithms not yet described in the text. The figure, if it is really needed, should be moved in the results section.

3.2 How success of the threshold based segmentation is evaluated ? It is a key point of the workflow.  

3.3 What is the means to perform lung segmentation if a local segmentation algorithm is used driven by user click? Both local segmentation algorithms are not affected by tissues surrounding the lungs.  

3.4.1 The paragraph is not clear and redundant. Authors state that “for each nodule, the average and standard deviation are calculated “, so why we need to “estimate the unknown average of a nodule” if the average value is known by the previous step? Equation (1) do not make sense. For a “ideal” nodule with no added noise (avg=start for the ideal nodule) we will have a totally wrong result applying equation (1). Is eq (1) related to the conversion from DICOM pixel values to HU units defined in the slope/intercept DICOM fields?  The conversion formula is perfectly know from DICOM. The second part is totally redundant. Authors use a very complicated description to define the use of the standard formula threshold = mean -2SD. Instead, would be of more interest to detail the SD estimation procedure. Figure 6 is redundant as well. Why we should call “features” pixel values? What neighbors shape was used in the region growing algorithm ?

3.4.3 Have authors evaluate the use of weighted loss function in training to address imbalance problem?

3.5.1 When morphological operation are performed? In all nodules or only nodules with over-segmentation? How “nosies removal” was performed? Labeling algorithm?

4 If we have two main stage why three experimental data set?

4.4.1 What author mean for “Normal” patients? Patients without nodules of patients with non invasive nodules? Patients with nodules are not normal.

4.4.2 Why lung segmentation is a “fundamental component” if a local segmentation algorithm is used in the following?

Table 1 is redundant, we do not need a “scholastic” explanation of confusion matrix concept. The same for figure 13 and others.

The paper should be revised for language quality

Author Response

The authors are grateful for the reviewers’ helpful comments that are valuable in improving this paper. We have revised the manuscript as follows.

Revision made in accordance with comments by Reviewer No.2

  1. The paper is very long and hard to follow, due to the redundance of information. Despite the length, some important detail are missed don’t’ allowing reproducibility of results. My advice is t dramatically shorten the paper attempting to focus on the introduced novelty and better detailing the proposed methodology.

Answer: Thanks for this comment. For this comment, we have deleted those the reviewer suggested, such as the original Subsection 3.1, Figure 6, Figure 8, Figure 13, Table 1 and so on. However, because some contents were added into this revision based on the other reviewers’ comments, the paper length was not shortened significantly.

  1. Authors seems to use three different data set to validate each step of the workflow. This kind of validation will no demonstrate the effectiveness of the whole workflow. Why do no test the whole algorithm on a single data set as the last one?

Answer: Thanks for this comment. Actually, there are only two datasets used in the experiments. One was from kaggle competitions and the other was from Departments of Diagnostic Radiology and Surgery, Kaohsiung Chang Gung Memorial Hospital (called KCGMH), Taiwan. The kaggle data was used for lung segmentation and KCGMH data was used for nodule segmentation and invasiveness recognition. The reason for why using two different sets can be answered by two points. First, the lung segmentation is the preprocessing of nodule segmentation but it needs much effort to mark the training lungs. To save the effort, we tried to use the existing kaggle data instead of KCGMH data for training a lung segmentation model. Because the final nodule segmentation was effective, we did not use the KCGMH data to train the lung segmentation model. That is, the lung segmentation was completed finally by the kaggle training model. Yet, it leaves a future issue to be investigated for the effectiveness if using the KCGMH data. Second, on the contrary, the kaggle data is not with the invasiveness information. Therefore, it cannot be used for invasiveness recognition. For these considerations, the kaggle data was used for lung segmentation, and the KCGMH data was used for nodule segmentation and invasiveness recognition. The related descriptions have been added into Subsection 4.1.1 (P. 10).

  1. Abstract: The abstract should resume the overall paper content, including the significant results.

Answer: Thanks for this comment. For this comment, we have modified the abstract to show the significant results. Please find below for a quick review.

The extensive experimental results on a real dataset reveal the proposed segmentation method performs better than the traditional segmentation ones, which can reach an average dice improvement of 392.3%. Also, the proposed ensemble classification model infers better performances than the compared method, which can reach a AUC (Area Under Curve) improvement of 5.3% and a specificity improvement of 14.3%. Moreover, in comparison with the models using imbalance data, the improvements of AUC and specificity can reach 10.4% and 33.3%, respectively. 

  1. Introduction: How invasive/non invasive nodules are classified in the clinical practice? By size ,signal or shape?

Answer: Thanks for this comment. In practical, irregular shape, solid component and tumor size are three important considerations for identifying the invasiveness. We have added this point into Section 1 (P. 2).

  1. I don’t see the need o this section. Authors mix the description of the implemented algorithm with discussion about advantages of the proposed implementation. I suggest to write a more rigorous method section with a objective description of the implemented workflow. Advantages of the methos should be described I the discussion section basing on obtained results.

Answer: Thanks for this comment. As suggested, we have deleted the original Subsection 3.1 and moved the examples to Subsections 3.3.2 (P. 6) and 4.2.4 (P. 13).

  1. Figure 2/3 anticipates results of algorithms not yet described in the text. The figure, if it is really needed, should be moved in the results section.

Answer: Thanks for this comment. For Figure 2, it was moved to Subsection 4.2.4 and renamed as Figure 13. For Figure 3, because it is an idea example for the revised Mask-RCNN, we moved it to Subsection 3.3.2 and renamed as Figure 4.

  1. 2 How success of the threshold based segmentation is evaluated ? It is a key point of the workflow.

Answer: Thanks for this comment. The unsuccess indicates that the thresholding-based morphology cannot find the nodules. For this comment, we modified the description in Subsection 3.1 (P. 5). Also, the related description has been shown in Subsection 3.4.2 (P. 9).

  1. 3 What is the means to perform lung segmentation if a local segmentation algorithm is used driven by user click? Both local segmentation algorithms are not affected by tissues surrounding the lungs.

Answer: Thanks for this comment. In the proposed method, the search space of thresholding-based segmentation is limited in the lung but that of Mask-RCNN is not. If the search space is not limited in the lung, the thresholding-based segmentation will connect the non-nodule tissues out of the lung. Figure 13 illustrates this idea and the related description is shown in Subsection 4.2.4 (P. 13).

  1. 4.1 The paragraph is not clear and redundant. Authors state that “for each nodule, the average and standard deviation are calculated “, so why we need to “estimate the unknown average of a nodule” if the average value is known by the previous step? Equation (1) do not make sense. For a “ideal” nodule with no added noise (avg=start for the ideal nodule) we will have a totally wrong result applying equation (1). Is eq (1) related to the conversion from DICOM pixel values to HU units defined in the slope/intercept DICOM fields?  The conversion formula is perfectly know from DICOM. The second part is totally redundant. Authors use a very complicated description to define the use of the standard formula threshold = mean -2SD. Instead, would be of more interest to detail the SD estimation procedure. Figure 6 is redundant as well. Why we should call “features” pixel values? What neighbors shape was used in the region growing algorithm?

Answer: Thanks for this comment. As suggested, we have deleted Figure 6.

In the training stage of thresholding-based segmentation, there are three outputs including Eq. (1), a multiple regression model and a threshold formula, and there are three inputs including ground truths of starts, averages and standard deviations. For Eq. (1), we need to calculate the true averages from the training data. By using the true starts and true averages, the coefficients of Eq. (1) can be approximated. The Eq. (1) is defined as:

avg01 * start  (1)

Next, by using the true starts, true averages and true standard deviations, a multiple regression model is constructed. Then, we can obtain Eq. (2), which is defined below.

std = Regression(start, avg)  (2)

By using start, Eq. (1) and Eq. (2), the Eq. (3) is derived for online estimating the threshold, which is defined as:

threshold = avg - a * std  (3)

In the online stage, the user click is the input. Next, the average will be estimated by Eq. (1). Then, the predicted average and the user click are input into the multiple regression model for estimating the standard deviation. Finally, the threshold is approximated by Eq. (3) using the avg and the std as the input.

In this paper, the thresholding-based segmentation is performed by the hu values. Therefore, a feature indicates a hu value in this method. Also, it uses eight neighbors surrounding the central cell as the connection shape. In the morphology, the kernel size is 3*3 and the number of iterations is 1. We have added the related descriptions into Subsection 3.4.1 (P. 8).

  1. 4.3 Have authors evaluate the use of weighted loss function in training to address imbalance problem?

Answer: Thanks for this comment. No, we did not. For this comment, we have added it into the future works (Section 6, P. 18). 

  1. 5.1 When morphological operation are performed? In all nodules or only nodules with over-segmentation? How “nosies removal” was performed? Labeling algorithm?

Answer: Thanks for this comment. In this paper, the morphological operations are performed for each nodule, including one erode and one dilate. After eroding, the original segmentation might be divided into several sub-segmentations. Then, the sub-segmentation including the user click will be kept. The others are deleted. For this concern, we have added the detailed descriptions into Subsection 3.4.1 (P. 8). 

  1. 4 If we have two main stage why three experimental data set?

Answer: Thanks for this comment. Actually, there are only two datasets used in the experiments. One was from kaggle competitions and the other was from Departments of Diagnostic Radiology and Surgery, Kaohsiung Chang Gung Memorial Hospital (called KCGMH), Taiwan. The kaggle data was used for lung segmentation and KCGMH data was used for nodule segmentation and invasiveness recognition. The reason for why using two different sets can be answered by two points. First, the lung segmentation is the preprocessing of nodule segmentation but it needs much effort to mark the training lungs. To save the effort, we tried to use the existing kaggle data instead of KCGMH data for training a lung segmentation model. Because the final nodule segmentation was effective, we did not use the KCGMH data to train the lung segmentation model. That is, the lung segmentation was completed finally by the kaggle training model. Yet, it leaves a future issue to be investigated for the effectiveness if using the KCGMH data. Second, on the contrary, the kaggle data is not with the invasiveness information. Therefore, it cannot be used for invasiveness recognition. For these considerations, the kaggle data was used for lung segmentation, and the KCGMH data was used for nodule segmentation and invasiveness recognition. The related descriptions have been added into Subsection 4.1.1 (P. 10). 

  1. 4.1 What author mean for “Normal” patients? Patients without nodules of patients with non invasive nodules? Patients with nodules are not normal.

Answer: Thanks for this comment. Yes, we agreed with this point and it is really our careless fault. For this comment, we modified “normal” into “non-invasive” in this revision (Subsection 4.1.1, P. 10). 

  1. 4.2 Why lung segmentation is a “fundamental component” if a local segmentation algorithm is used in the following?

Answer: Thanks for this comment. According to Figure 2, the lung segmentation is the preprocessing component of the thresholding-based segmentation. That is, the search space for thresholding-based segmentation is limited in the lung. If it cannot find the nodule, the segmentation will be switched to Mask-RCNN without lung segmentation. The related description is shown in Subsections 3.1 (P. 4), 3.4.1 (P. 8) and 3.4.2 (P. 9). 

  1. Table 1 is redundant, we do not need a “scholastic” explanation of confusion matrix concept. The same for figure 13 and others.

Answer: Thanks for this comment. For this concern, we deleted Table 1 and Figure 13. Also, the related descriptions are also deleted.

Reviewer 3 Report

This paper presents the recognition of the invasiveness of imbalanced data using semi-automatic lung nodule segmentation.

 The article is an interesting approach to the analysis of medical images. Nevertheless, an important element in learning-based methods is the training and validation set. The effectiveness of forecasting and the thesis depends largely on the selection of criteria, the amount of data, and the appropriate selection of data. The universality of the algorithm, therefore, has certain limitations.

 Minor remarks:

1) I propose to describe in more detail on what basis and what results from the scope of selection of input data for the research problem and to what extent the algorithm's effectiveness depends on the amount and type of actual data.

2) The authors could refer more broadly to other teaching methods to analyse a similar research problem and justify the solution's choice.

Author Response

The authors are grateful for the reviewers’ helpful comments that are valuable in improving this paper. We have revised the manuscript as follows.

Revision made in accordance with comments by Reviewer No.3

  1. I propose to describe in more detail on what basis and what results from the scope of selection of input data for the research problem and to what extent the algorithm's effectiveness depends on the amount and type of actual data.

Answer: Thanks for this comment. In the experiments, the data is composed of two sets. One was from kaggle competitions and the other was from Departments of Diagnostic Radiology and Surgery, Kaohsiung Chang Gung Memorial Hospital (called KCGMH), Taiwan. The kaggle data was used for lung segmentation and KCGMH data was used for nodule segmentation and invasiveness recognition. The reason for why using two different sets can be answered by two points. First, the lung segmentation is the preprocessing of nodule segmentation but it needs much effort to mark the training lungs. To save the effort, we tried to use the existing kaggle data instead of KCGMH data for training a lung segmentation model. Because the final nodule segmentation was effective, we did not use the KCGMH data to train the lung segmentation model. That is, the lung segmentation was completed finally by the kaggle training model. Yet, it leaves a future issue to be investigated for the effectiveness if using the KCGMH data. Second, on the contrary, the kaggle data is not with the invasiveness information. Therefore, it cannot be used for invasiveness recognition. For these considerations, the kaggle data was used for lung segmentation, and the KCGMH data was used for nodule segmentation and invasiveness recognition. The related descriptions have been added into Subsection 4.1.1 (P. 10).  

The results of evaluating the proposed methods on different datasets are shown in Subsections 4.2 and 4.3. For lung segmentation, the result is shown in Subsection 4.2.1. For nodule segmentation, the result is shown in Subsections 4.2.2-4.2.4. For invasiveness recognition, the result is shown in Subsection 4.3.

For scalability concerns, we also conducted two more evaluations. One is for nodule segmentation and the other is for invasiveness recognition. Although the larger training data sizes for all measures achieve the better results, the differences are not significant. It delivers an aspect that the proposed method is not very sensitive to the training data size. The related evaluation results are shown in Figures 21 and 22 (Subsection 4.4-IV, P. 17- P. 18). Also, it is one of the research limitations stated in Section 5 (P. 18).

  1. The authors could refer more broadly to other teaching methods to analyse a similar research problem and justify the solution's choice.

Answer: Thanks for this comment. Because this paper contains two parts, this question can be answered by two aspects, namely nodule segmentation and invasiveness recognition.

For nodule segmentation, the recent solution is the fully-automated segmentation searching the nodules by scanning the whole images. In contrast, this paper proposes a semi-automated segmentation with the click information. Therefore, it is more effective because the search space is narrowed into the areas surround the clicked pixel. To clarify the effectiveness difference, we conducted a comparison between the proposed method and fully-automated ones. For this issue, we added one more compared method in this revision. The related experimental analysis is shown in Subsection 4.4-III (Figure 20, P.17).

In addition to nodule segmentation, another answer is for invasiveness recognition. In the experiments, we tried our best to test considerable well-known methods widely used in the field of Bioinformatics, including 4 feature-selections and 6 classifiers. For this concern, we added one more feature selection method Pearson Correlation into the experiments (Figure 17). The related analysis was shown in Subsection 4.3.1 (P. 14). In overall, the effectiveness differences among the compared feature-selection methods are not obvious. For this comment, it was also added as a future work (Section 6, P. 19) for seeking better solutions for invasiveness recognition.

Round 2

Reviewer 2 Report

The paper was improved in several aspects and the need for different datasets is now more clear.

There are still some issues;

I still do not understand the utility of equation (1). Real CT values can be calculated from HU value by Rescale Intercept and RescaleSlope attributes in DICOM by the formula:

Value = (HU- Rescale Intercept)/ RescaleSlope

So b0 = -RI/RS  and b1=1/RS, and there is no need to evaluate b0/b1 from experiments. The use of equation (1) approach would lead to the need to calibrate the procedure from each new dataset  

Table 1 add SD of dice value across experiments

Institutional Review Board Statement, Informed Consent Statement, and Data Availability Statement were not compiled. Are datasets used in the study accessible to allow the assessment of study reproducibility?

There are several grammatical errors that should be corrected, for instance:

Line 213: Equitation -> equation

Equation 4: prdicted -> predicted   tuth -> truth

Equations 4-10 be consistent with the use of uppercase/fonts

Author Response

The authors are grateful for the reviewers’ helpful comments that are valuable in improving this paper. We have revised the manuscript as follows.

Revision made in accordance with comments by Reviewer No.2

  1. I still do not understand the utility of equation (1). Real CT values can be calculated from HU value by Rescale Intercept and RescaleSlope attributes in DICOM by the formula:

Value = (HU- Rescale Intercept)/ RescaleSlope

So b0 = -RI/RS  and b1=1/RS, and there is no need to evaluate b0/b1 from experiments. The use of equation (1) approach would lead to the need to calibrate the procedure from each new dataset   

Answer: Thanks for this comment. We are sorry that the related description is still unclear.

In the comment, the suggested formula is used for calculating the hu values from CT stored values by the Rescale Intercept and RescaleSlope. In contrast, the proposed Eq. (2) is used for online calculating the average of unknown nodule hu values. (In this revision, the original Eq. (1) was renamed as Eq. (2).) The utilities of them are different. Therefore, in the proposed method, the b0 and b1 are approximated by a simple linear regression, different from the ones suggested. To clarify this issue, we conducted a more experiment for the effectiveness comparison. Based on the suggested coefficients of b0 (b0 = -RI/RS) and b1 (b1=1/RS), all testing dices of the experimental data are zero. This is because the threshold is too high to partition the nodule. The following Table A depicts the comparison examples.

Table A. Comparison examples for dices derived by proposed threshold and reviewer suggested threshold, respectively.

Proposed

Threshold

Dice

by Proposed Threshold

Reviewer

Suggested Threshold

Dice

by Reviewer

Suggested Threshold

Nodule 1

-495.714

0.914

1208.792

0

Nodule 2

-629.217

0.86

919.158

0

Nodule 3

-688.47

0.765

919.158

0

Nodule 4

-753.657

0.726

620.42

0

Nodule 5

-797.728

0.943

644.419

0

For this comment, we added the commented formula into the second step of the training procedure (Subsection 3.3.1, P. 6). Also, the related descriptions (Subsection 3.3.1, P. 6) were modified to make this idea clearer. Note that, the above Table A is just an illustrative example. Because of the paper length concern, Table A was not added into the revision.

  1. Table 1 add SD of dice value across experiments

Answer: Thanks for this comment. We have added the standard deviations into Table 1 and added the related descriptions into Subsection 4.2.1 (P. 11).

  1. Institutional Review Board Statement, Informed Consent Statement, and Data Availability Statement were not compiled. Are datasets used in the study accessible to allow the assessment of study reproducibility?

Answer: Thanks for this comment. No, the experimental data will not be available because of the ethical concerns. The related statements for Institutional Review Board Statement, Informed Consent Statement and Data Availability Statement have been added into the revision (P. 19).

  1. There are several grammatical errors that should be corrected, for instance
  • Line 213: Equitation -> equation
  • Equation 4: prdicted -> predicted   tuth -> truth
  • Equations 4-10 be consistent with the use of uppercase/fonts

Answer: Thanks for this comment. The careless errors have been corrected. Then, we conducted a careful check again.

Round 3

Reviewer 2 Report

I don't have further comments

English quality acceptable